# Exploring the Potential of Roadside Plantation for Carbon Sequestration Using Simulation in Southern Quebec, Canada

**Nour Srour [1,*], Evelyne Thiffault [1]** and **Jean-François Boucher [2]**

1   Research Center on Renewable Materials, Department of Wood and Forest Sciences, Université Laval, Quebec City, QC G1V 0A6, Canada; evelyne.thiffault@sbf.ulaval.ca
2   Department of Fundamentals Sciences, Université du Québec à Chicoutimi, Chicoutimi, QC G7H 2B1, Canada; jean-francois_boucher@uqac.ca
*   Correspondence: nour.srour.1@ulaval.ca

**Abstract:** Afforestation of urban lands can play an important role in increasing carbon sequestration and mitigating the effects of climate change. This study aimed to evaluate the potential for carbon sequestration and storage in plantations on roadsides in southern Quebec, Canada. We used the Carbon Budget Model of the Canadian Forester Sector 3 (CBM-CFS3) to simulate the carbon sequestration capacity over 100 years of plantations established following afforestation with different species mixtures based on local needs and aspirations. We then compared the carbon sequestration potential of simulated plantations with the carbon storage of natural vegetation of reference roadsides with different histories of land use. Our findings suggested that plantations on roadsides subjected to high anthropogenic pressure, such as road rights-of-way, may provide carbon sequestration benefits relative to baseline conditions (i.e., no plantation). For instance, 15 years after afforestation, the additional carbon sequestration potential of plantations on road rights-of-way varied between 25 and 32 Mg ha$^{-1}$, depending on the afforestation scenario. However, allowing roadsides classified as abandoned agricultural lands to undergo natural succession could promote higher carbon storage on roadsides than planting, irrespective of species mixtures. Our results indicated that the carbon storage of vegetation resulting from the abandonment of agriculture 35 to 45 years ago showed a range of 260 to 290 Mg ha$^{-1}$, which exceeded the carbon stocks predicted with afforestation models for 60 to 84 years after planting. Indeed, reference roadsides used for agriculture in the past, but that have otherwise not been subjected to other anthropogenic degradation, appeared to naturally evolve toward forest vegetation with higher carbon stocks than simulated plantations.

**Keywords:** afforestation; carbon sequestration; CBM-CFS3; roadsides; species mixtures; natural succession





## 1. Introduction

The increase in greenhouse gas (GHG) emissions to the atmosphere caused by human activities has been identified as the main reason for rapid climate change in recent decades, with carbon dioxide ($CO_2$) as the most abundant GHG emitted to the atmosphere [1]. To limit global warming to 1.5 °C above the pre-industrial level, mitigation activities should be considered to decrease $CO_2$ emissions and achieve net-zero emissions by 2050 through increased absorption by carbon sinks [1].

Afforestation and management of currently unforested, degraded, or urban lands are considered important practices to increase terrestrial carbon sinks [1,2]. The potential for carbon sequestration and climate change mitigation of afforestation of these lands have been studied in various jurisdictions, notably, in Canada, as part of a portfolio of nature-based solutions [3–5]. Lands alongside road networks are examples of under-utilized areas that could contribute to increased carbon storage through afforestation [6–10]. These strips of land, which extend from the edge of the road to the surrounding landscape, are typically managed by government services at the city, province, or country level. Previous

studies have shown the success of active management programs dedicated to roadsides. For instance, a tree planting initiative along highway borders in China sequestered 4.68 Mg C between the 1980s and 2005 [9]. Similarly, in the United States of America, Ament and Begley [11] found that tree plantation along 485,255 km of roads had the potential to sequester 8 Mg C per year. In Brazil, afforestation of areas along highways could sequester up to 655 Mg $CO_2$ (179 Mg C) per kilometer over ten years [7]. In Bangladesh, the average carbon content of roadside plantations is 192.80 Mg ha$^{-1}$, with 86% of carbon stored aboveground and 14% belowground [8].

As part of its climate policy, the government of Quebec, in eastern Canada, is considering the opportunity to manage and protect existing ecosystems located on roadsides and afforest currently unforested roadsides. Across Quebec, local stakeholders and land managers are involved in selecting potential sites and species to ensure plantations reflect local needs, constraints, and aspirations for increasing the resilience of regional landscapes in a changing climate. Local communities play an essential role in decision-making, promoting participation, and providing a sense of global responsibility [12]. Additionally, they may bring traditional expertise and historical knowledge regarding their environment [13] to aid in identifying suitable species for plantation. This contribution can enhance the sustainability of afforestation projects and foster successful climate change mitigation [14]. Therefore, in Southern Quebec, pilot plantations have recently been established on small roadside areas (alongside control, non-planted plots) in collaboration with local stakeholders to test and improve planting and monitoring practices and generate empirical data about ecosystem carbon stock evolution [15]. The pilot plantations and control plots will provide, over the long term, the necessary data for evaluating the efficiency of roadside planting as a carbon sequestration measure.

The carbon sequestration benefit of such afforestation activities can be estimated based on the difference between the afforestation scenario and a baseline scenario representing the business-as-usual scenario (i.e., no plantation on unforested lands) [16]. In the absence of long-term empirical data, as is the case for the pilot plantations in Quebec, the Carbon Budget Model of the Canadian Forest Sector 3 (CBM-CFS3) [17] is widely used in Canada to simulate forest carbon dynamics at the stand or landscape scales [18]; it has been used to estimate the potential of afforestation in a variety of contexts [3,5,19]. However, while CBM-CFS3 is well adapted to the simulation of tree-dominated ecosystems such as plantations, it is less suited for representing herbaceous and shrub vegetation, which can dominate in the early stages of natural succession [20], or for simulating structurally complex stands. For instance, after agriculture abandonment of a given land, the typical approach to managing the area is allowing nature to take its course [21], which can result in carbon accumulation in both aboveground and belowground pools. Indeed, recent studies in Quebec suggest that on previously unforested sites such as abandoned agricultural lands and in the absence of active tree plantation, natural succession can evolve from herb- and shrub-dominated stages to multi-layered stands and sequester and store important quantities of carbon in vegetation biomass and soils without human intervention [15]. The level of C stocks in such naturally established stands can sometimes be similar to that obtained with tree plantation, at least for the first decades [20,22]. Yet, the combination of active management practices (human intervention) and passive management practices (without human intervention) could increase ecosystem carbon sequestration [21]. Comparing carbon sequestration of plantation scenarios to baseline conditions (i.e., the natural evolution of unforested areas) is therefore crucial to evaluating the opportunity to invest in plantations as part of a climate mitigation plan.

This study evaluated the long-term carbon sequestration and storage capacity of plantations established on roadsides, using southern Quebec as a regional case study. In the absence of long-term field data, the CBM-CFS3 model was used to simulate carbon dynamics following tree planting on different roadsides using various species mixtures reflecting local needs and aspirations. We then compared the carbon sequestration potential of plantations based on different afforestation scenarios with the carbon storage of existing

natural vegetation found on roadsides as reference sites, which served as a validation of the simulation results and an evaluation of plantation performance. We considered the following ecosystem carbon pools: live biomass (including above and belowground biomass), dead organic matter (including snags, dead woody debris, litter, and forest floor), and soil down to a 55 cm depth.

## 2. Materials and Methods

### 2.1. Study Area

This study was conducted in 8 sectors ranging in size from 5 to 100 ha and distributed in three administrative regions of southern Québec: Mauricie (4 sectors), Montréal (2 sectors), and Montérégie (2 sectors) (Figure 1). These regions are located in the sugar maple–bitternut hickory and the sugar maple–basswood bioclimatic domains [23]. The sugar maple–bitternut hickory domain has an average altitude of 57 m, an annual temperature average of 6.2 degrees Celsius, a growth season that spans 187 days, and an annual precipitation of 1005 mm [24]. It is the most densely populated area of Quebec; human intervention has, therefore, significantly impacted the vegetation of this area over the past two centuries. The sugar maple–basswood bioclimatic domain is situated at an average altitude of 83 m, with an average annual temperature of 5.1 degrees Celsius, a growing season that lasts 178 days, and a total annual precipitation of 1095 mm [24]. Agricultural lands are abundant in this region, particularly in areas with clay deposits [24].

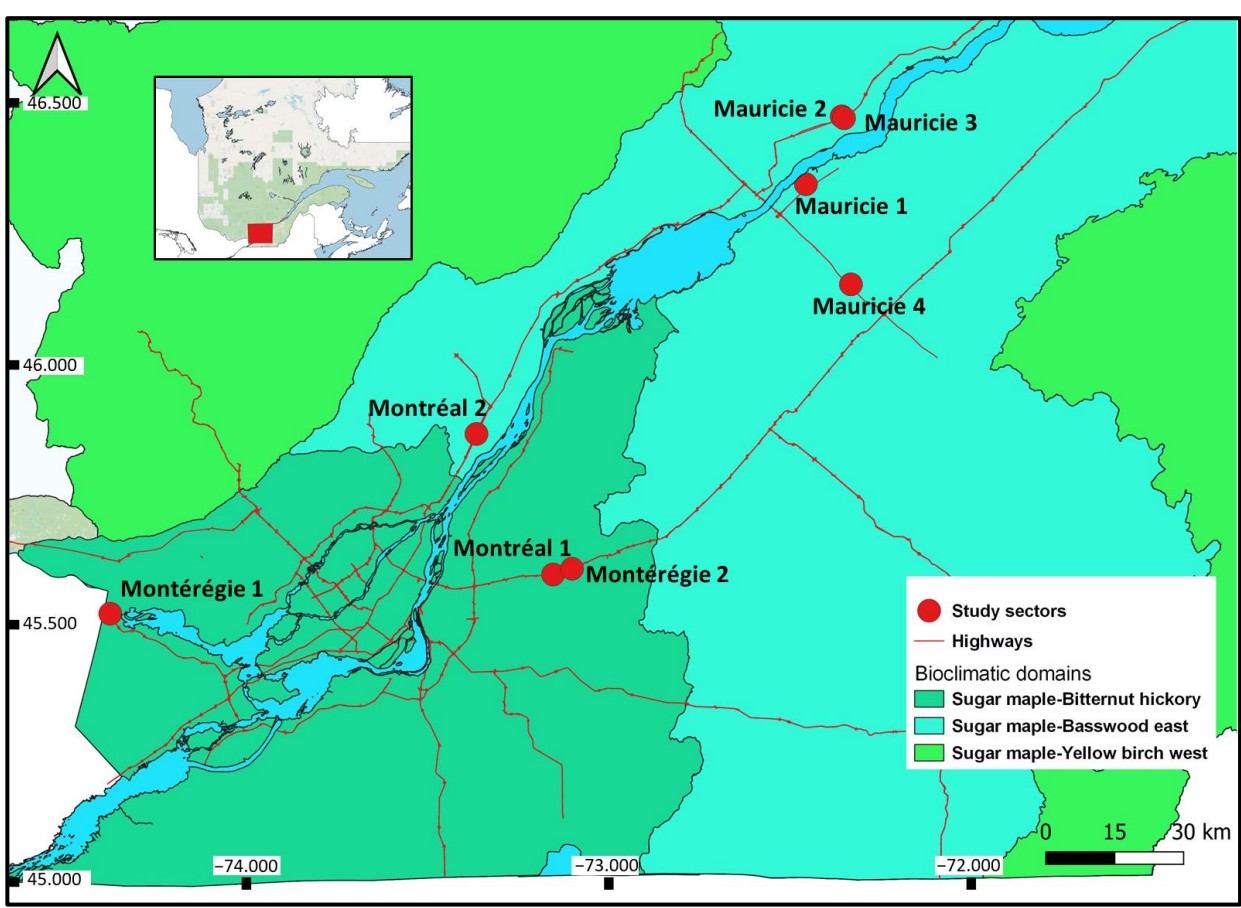

**Figure 1.** Distribution of the roadside study sectors in the Mauricie, Montréal, and Montérégie regions in Quebec province, Canada.

### 2.2. Reference Roadsides

To provide a form of validation for the simulated capacity of roadsides to sustain vegetation growth, and to evaluate the potential of carbon sequestration by planting, carbon

stocks of 52 reference roadsides were used as baseline comparisons for the simulated results of the tree planting scenarios. Reference sites were those selected and documented in Srour et al. [15]; these reference roadsides were not subjected to active plantation. Briefly, the 52 reference sites were selected based on the availability of roadsides in the 8 different sectors of the three studied regions (Figure 1). We listed all available roadsides that met specific size criteria (300 m$^2$) and classified them based on their vegetation type and assemblage. A gradient of visually contrasting vegetation was then selected by randomly choosing roadsides within each combination of vegetation type/assemblage of each region. Fifty-two roadsides comprising different types, such as highway medians, highway rights-of-way, local and regional collectors, and arterial roads, were selected, with one inventory plot established on each distinct roadside. Soil samples were collected from the first 15 cm of the mineral horizon on each roadside (see below for the sampling procedure) and tested for soil texture using the Bouyoucos method [25]. The soils of various particle size distributions were more or less randomly distributed among the 52 selected roadsides. Clay percentage varied between 4.0% and 51.8%, 2.0% and 58.0%, and 14.0% and 38.2% for roadsides located in the Mauricie, Montréal, Montérégie regions, respectively, while sand percentage varied between 11.7% and 88.0%, 19.1% and 87.0%, and 25.6% and 56.1%.

The sampling procedure of carbon stocks was based on Canada's National Forest Inventory ground sampling guidelines [26]. The detailed sampling procedure of each carbon reservoir in these sites was described in Srour et al. [15]. Briefly, one circular main plot of 400 m$^2$ (28 roadsides) or 200 m$^2$ (24 roadsides) (depending on the density and diameter at breast height (DBH) of trees) were established in each of the selected roadsides (fifty-two plots in total). Large trees and snags with a height of at least 1.3 m and diameter at breast height DBH > 9 cm were sampled on the area of each main plot. Small trees and shrubs were inventoried within one plot with a 3.98 m radius (50 m$^2$) located at the center of each main plot. Understory vegetation, including herbaceous or woody plants with a diameter at stump height (DHS) < 1 cm, was sampled within four 1 m$^2$ quadrats located within the main plot. Woody debris was sampled along two perpendicular 10 m transects, and the diameter, species, genus, and decay class (i.e., 5 classes; class 1: wood texture is intact and hard; class 2: wood texture is intact and partly decaying; class 3: wood texture is hard with large pieces and partly decaying; class 4: wood is found in small and block pieces; class 5: wood is found in small pieces and soft portions) were noted. The soil was sampled at two soil sampling stations located at the border of each plot. When present, the L and FH horizons were separately collected using a 20 cm × 20 cm template, and their depth was measured. The mineral horizons were sampled using a metal core with an internal diameter of 5 cm at three depths: 0–15 cm, 15–35 cm, and 35–55 cm.

L horizons and understory vegetation samples were oven-dried at 65 °C for constant mass (48–72 h) and weighed. FH and mineral soil samples were analyzed in the laboratory to determine their carbon content. Samples were air-dried for 7 days, sieved, and weighed before and after sieving. The bulk density of soil samples was calculated as described in Federer et al. [27]. The carbon concentration of soil samples was measured using a Leco CNS elemental analyzer (LECO Corporation, St Joseph, MI, USA).

To determine the amount of carbon stored in each plot, we estimated the carbon stock in vegetation (above- and belowground), dead organic matter (i.e., litter and woody debris), and soil. The biomass and carbon stock of aboveground vegetation were estimated using allometric equations from the scientific literature [28–34] based on either DBH or DHS. The carbon stock of aboveground biomass of vegetation was multiplied by 1.256 to consider the carbon content of the root system [35]. The carbon content of L horizon samples and understory vegetation was estimated based on their oven-dry mass. Using the procedures outlined in the National Forest Inventory [36], we calculated the woody debris biomass based on each decomposition class and species. To convert the biomass into carbon stock (in Mg ha$^{-1}$), we multiplied by the conversion factor of 0.5 Mg of C/Mg of oven-dry biomass, as recommended by Penman et al. [16]. All carbon stock data were then converted into megagrammes per hectare (Mg ha$^{-1}$).

We used historical maps to assess the evolution of the vegetation of the reference roadsides (i.e., sites for which no active plantation was performed) to provide some background about the potential trajectory of roadsides in the absence of active management for carbon sequestration. The ecoforest map of Quebec, updated every ten years using aerial photos, provided data on the historical development of reference roadsides from 1975 (the oldest available map for the province) to 2015 (the most recent inventory) [23]. The evolution of vegetation on roadsides was assessed using the land-use classification obtained from the ecoforest maps. Roadsides were classified according to four land-use types: agricultural land, abandoned agricultural land, right-of-way, and forest; the land-use type of each reference roadside in 1975, 1985, 1995, 2005, and 2015 was hence recorded. For each region, roadsides with a similar land-use evolution pattern were grouped, and the average ecosystem carbon stocks for a given group were calculated and used as validation and comparisons for the simulated plantations of this region. As the data suggested that most roadsides appeared to have been used as agricultural lands at some point in the past, the approximate time since the abandonment of agriculture (and likely the onset of natural succession) was estimated based on the maps, when relevant. For sites that were continuously classified as forests since the earliest inventory (1975), a minimum age of 45 years was estimated. Overall, we sampled 34 roadsides currently classified as forest, 8 roadsides currently classified as abandoned agricultural land, 8 roadsides currently classified as rights-of-way, and 2 roadsides currently classified as agricultural land.

### 2.3. Afforestation Scenarios

The simulation of carbon dynamics was carried out for four scenarios of afforestation of roadsides for each of the three regions. First, potential roadside candidates for afforestation were identified by local land managers among the 52 reference roadsides (see above) [37]. Candidate sites were selected based on their ease of access for future planting and maintenance crews and the current absence of abundant tree vegetation. The candidate sites for afforestation in the Mauricie region were classified under the sugar maple–basswood bioclimatic domain and were predominantly clayey or silty with imperfect drainage. In Montréal, the candidate sites were situated in the sugar maple–bitternut hickory domain and were characterized as heavy clay with moderate or imperfect drainage. In Montérégie, the candidate sites were located in the sugar maple–bitternut hickory domain and were sandy and well-drained (Figure 1). For the sake of simulation, the ecological characteristics of the candidate sites were averaged for a given region to determine one typical roadside per region for which afforestation simulations would be performed. This ensured that the site conditions used for the simulation would be broadly representative of the region and similar to the reference roadsides used for validation and comparison.

For each region, the mixtures of planted species for four different afforestation scenarios were determined according to the needs and preferences of local stakeholders and land managers and the characteristics of the simulated roadside. The afforestation scenarios were: Standard, Diversified, Limited Maintenance, and Assisted Migration. The exact mixtures of species for each scenario varied according to the study region (Table 1). The Standard scenario represented a mixture of equal parts of locally relevant species. The Limited Maintenance scenario considered species that can acclimate to competitive, low-maintenance, and open environments, i.e., species known to be suitable for site colonization. The Diversified scenario was characterized by an abundance of long-lived broadleaved species, with conifers and other companion species; this mix of species was considered a favourable scenario for carbon sequestration. The Assisted Migration scenario integrated species that are not currently present in the bioclimatic domain of the study sites but are abundant in more southern regions.

**Table 1.** Species composition and corresponding percentage of stem numbers planted per hectare for the afforestation scenarios. The percentage of planted species is based on a density of 800 and 2000 stems per hectare for hardwood and conifer species, respectively.

| Afforestation Scenarios | Mauricie | | Montréal | | Montérégie | |
|---|---|---|---|---|---|---|
| | Percentage of Planted Stems per Hectare (%) | Planted Species | Percentage of Planted Stems per Hectare (%) | Planted Species | Percentage of Planted Stems per Hectare (%) | Planted Species |
| Standard | 14.3 | Hybrid poplar | 14.3 | Hybrid poplar | 14.3 | Hybrid poplar |
| | 7.1 | *Picea glauca* | 7.1 | *Picea glauca* | 7.1 | *Picea glauca* |
| | 7.1 | *Picea abies* | 7.1 | *Picea abies* | 7.1 | *Picea abies* |
| | 14.3 | *Thuja occidentalis* | 14.3 | *Thuja occidentalis* | 14.3 | *Thuja occidentalis* |
| | 14.3 | *Betula papyrifera* | 14.3 | *Betula papyrifera* | 14.3 | *Betula papyrifera* |
| | 14.3 | *Acer saccharinum* | 14.3 | *Acer saccharinum* | 14.3 | *Prunus serotina* |
| | 14.3 | *Quercus macrocarpa* | 14.3 | *Quercus macrocarpa* | 14.3 | *Quercus rubra* |
| | 14.3 | *Acer rubrum* | 14.3 | *Acer rubrum* | 14.3 | *Acer rubrum* |
| Limited Maintenance | 8.3 | *Picea glauca* | 8.3 | *Picea glauca* | 8.3 | *Picea glauca* |
| | 8.3 | *Picea abies* | 8.3 | *Picea abies* | 8.3 | *Pinus resinosa* |
| | 8.3 | *Larix laricina* | 8.3 | *Larix laricina* | 8.3 | *Carpinus caroliniana* |
| | 12 | *Betula papyrifera* | 12 | *Betula papyrifera* | 12 | *Betula papyrifera* |
| | 13 | *Populus balsamifera* L. | 13 | *Populus balsamifera* L. | 13 | *Populus deltoides* |
| | 25 | *Quercus macrocarpa* | 25 | *Quercus macrocarpa* | 25 | *Quercus rubra* |
| | 25 | *Acer rubrum* | 25 | *Acer rubrum* | 25 | *Acer rubrum* |
| Diversified | 25 | Hybrid poplar | 25 | Hybrid poplar | 25 | Hybrid poplar |
| | 3.75 | *Picea glauca* | 3.75 | *Picea glauca* | 3.75 | *Picea glauca* |
| | 3.75 | *Picea abies* | 3.75 | *Picea abies* | 3.75 | *Picea abies* |
| | 2.5 | *Thuja occidentalis* | 2.5 | *Thuja occidentalis* | 2.5 | *Thuja occidentalis* |
| | 2.5 | *Betula papyrifera* | 2.5 | *Betula papyrifera* | 2.5 | *Betula papyrifera* |
| | 2.5 | *Acer saccharinum* | 2.5 | *Acer saccharinum* | 2.5 | *Prunus serotina* |
| | 2.5 | *Acer pensylvanicum* | 5 | *Carpinus caroliniana* | 2.5 | *Acer pensylvanicum* |
| | 2.5 | *Carpinus caroliniana* | | | 2.5 | *Ostrya virginiana* |
| | 20 | *Quercus macrocarpa* | 20 | *Quercus macrocarpa* | 20 | *Quercus macrocarpa* |
| | 20 | *Acer rubrum* | 20 | *Acer rubrum* | 20 | *Acer rubrum* |
| | 7.5 | *Tilia americana* | 3 | *Tilia americana* | 5 | *Pinus strobus* |
| | | | 3 | Hybrid elm | 5 | *Pinus rigida* |
| | | | 3 | *Acer nigrum* | | |
| | 7.5 | Hybrid elm | 3 | *Celtis occidentalis* | 5 | *Tsuga canadensis* |
| | | | 3 | *Quercus bicolor* | | |
| Assisted Migration | 25 | Hybrid poplar | 25 | Hybrid poplar | 25 | Hybrid poplar |
| | 3.75 | *Picea glauca* | 3.75 | *Picea glauca* | 3.75 | *Picea glauca* |
| | 3.75 | *Picea abies* | 3.75 | *Picea abies* | 3.75 | *Picea abies* |
| | 2.5 | *Thuja occidentalis* | 2.5 | *Thuja occidentalis* | 2.5 | *Thuja occidentalis* |
| | 2.5 | *Betula papyrifera* | 2.5 | *Betula papyrifera* | 2.5 | *Betula papyrifera* |
| | 2.5 | *Acer saccharinum* | 2.5 | *Acer saccharinum* | 2.5 | *Prunus serotina* |

**Table 1.** *Cont.*

| Afforestation Scenarios | Mauricie | | Montréal | | Montérégie | |
|---|---|---|---|---|---|---|
| | Percentage of Planted Stems per Hectare (%) | Planted Species | Percentage of Planted Stems per Hectare (%) | Planted Species | Percentage of Planted Stems per Hectare (%) | Planted Species |
| | 2.5 | *Acer pensylvanicum* | 5 | *Carpinus caroliniana* | 2.5 | *Acer pensylvanicum* |
| | 2.5 | *Carpinus caroliniana* | | | 2.5 | *Ostrya virginiana* |
| | 20 | *Quercus macrocarpa* | 20 | *Quercus macrocarpa* | 20 | *Quercus macrocarpa* |
| | 20 | *Acer rubrum* | 20 | *Acer rubrum* | 20 | *Acer rubrum* |
| | 3.75 | *Acer nigrum* | 5 | *Platanus occidentalis* | 3.75 | *Pseudotsuga menziesii* |
| | 3.75 | *Celtis occidentalis* | 5 | *Quercus palustris* | 3.75 | *Quercus velutina* |
| | 3.75 | *Quercus bicolor* | 5 | *Gleditsia triacanthos* | 3.75 | *Quercus coccinea* |
| | 3.75 | *Quercus palustris* | | | 3.75 | *Gleditsia triacanthos* |

### 2.4. Modeling Framework

CBM-CFS3 was used to simulate ecosystem carbon fluxes of the various afforestation scenarios at the scale of 1 hectare of roadside (to facilitate comparisons between scenarios and regions and with reference roadsides) over a simulation period of 100 years following afforestation. CBM-CFS3 was developed by the Canadian Forest Service to simulate the change in carbon stocks in the forest carbon pools recognized by the Intergovernmental Panel on Climate Change (IPCC), i.e., aboveground and underground biomass, litter, dead wood, and soil organic carbon. C stocks in dead wood and soils are further broken down into pools with different cycling rates. The model is used in Canada for the reporting of GHG emissions from the forest sector and has also been adapted and used in other countries, including Mexico [38] and countries across Europe [39]. The primary data input required for CBM-CFS3 is the stand growth curves for the simulated species (merchantable volume as a function of stand age) [17]. CBM-CFS3 converts the merchantable volume curves into biomass using specific expansion factors (Boudewyn et al. [40] for Canadian species). It then uses a disturbance matrix that describes the carbon transfers between forest pools and between the ecosystem and the atmosphere, according to the simulated natural and anthropogenic disturbances [17]. Initial carbon stocks can be estimated by simulating recurring wildfires until C stocks in slow-cycling dead organic matter pools have stabilized; alternatively, field data can also be used to initialize carbon stock values at the beginning of the simulation.

At the time of this study, soil ploughing had already been performed in the summer of 2022 before tree planting on a limited number of candidate roadsides for afforestation (see Section 3.2 for a description of candidate roadsides) (4 roadsides located in the Montérégie and Mauricie regions). On each of these candidate roadsides, an area was ploughed. A 3.99 m circular plot was established on each ploughed area (i.e., 4 treated plots). Each treated plot was paired with one control plot established in an untouched area to form four pairs of control and treated plots. Volumetric soil sampling down to a 55 cm depth, based on the National Forest Inventory ground plot protocol [26], was performed in four soil stations within each of the ploughed and control plots, and soil carbon concentration was determined for the soil samples using a Leco CNS elemental analyzer (LECO Corporation, MI, USA).

Soil carbon stocks at Time 0 for each simulated roadside were set in CBM-CFS3 based on the average values measured on candidate roadsides of each region. Initial soil carbon stocks (before ploughing) were therefore set at 99 (SE: ±9.72), 97 (SE: ±10.49), and 95 (SE: ±11.08) megagrammes of carbon per hectare (Mg C ha$^{-1}$) for roadsides in

Montréal, Mauricie, and Montérégie, respectively. Short-term soil carbon losses attributed to soil ploughing were estimated by comparing soil carbon content between control vs. ploughed plots for the 4 pairs of plots and averaging the difference over the 4 pairs. The calculated average carbon loss was 10 (SE: ±23.95) Mg C ha$^{-1}$; this loss was included in the simulations.

CBM-CFS3 can only simulate two species at a time in a given stand (one deciduous and one coniferous species), limiting its simulation capacity for complex species mixtures. Therefore, we simulated 1 ha sub-plots planted with each species (monospecific stands) present in the scenarios. We then used the model outputs for individual species to compile weighted averages of carbon stocks from monospecific stands based on the area-based proportions of each species in the various mixtures of scenarios and regions described in Table 1.

To simulate the evolution of carbon stocks for sub-plots of individual species, yield tables of merchantable volume (m$^3$ ha$^{-1}$) were used for each of them (Table 2); we selected stand site index (SI) values that were relatively conservative, according to the literature (Table 2), and based on 800 and 2000 trees ha$^{-1}$ for hardwood and conifer species, respectively. The site index refers to the average height of the most dominant trees in a particular stand at a specific reference age. In natural stands, this reference age is 50 years, while for planted stands, it is usually 25 years [41]. We also tested the impact of the species growth curve on the evolution of carbon stocks by increasing the values of the growth curves for simulated species by 5% and 10%.

*2.5. Data Analysis*

The visual representations of the results were created using the *ggplot2* package [42] in the R software package 4.1.2 [43]. Specifically, we used the *geom line* and the *geom point* functions to represent the simulation results and the carbon stocks in roadside references, respectively.

**Table 2.** Site index (SI) and growth curves used to simulate the carbon stock evolution of individual planted species over 100 years.

| Afforestation Species | Site Index (SI) (m) | Reference for the Growth Curve |
|---|---|---|
| Hybrid poplar | SI—Hardwood [44] | [44] |
| *Picea glauca* | SI 11 at 25 years [45] | [45] |
| *Picea abies* | SI 10 at 25 years [46] | [47] |
| *Thuja occidentalis* | SI 9 at 50 years [48] | [44] |
| *Betula papyrifera* | SI 20 at 50 years [41] | [44] |
| *Acer saccharinum* | SI 10 at 50 years (lowest SI) | [49] |
| *Acer pensylvanicum* | SI 10 at 50 years (lowest SI) | [49] |
| *Carpinus caroliniana* | SI 10 at 50 years (lowest SI) | [49] |
| *Quercus macrocarpa* | SI 10 at 50 years (lowest SI) | [49] |
| *Acer rubrum* | SI 17 at 50 years (lowest SI) | [50] |
| *Tilia americana* | SI 10 at 50 years (lowest SI) | [49] |
| Hybrid elm | SI 10 at 50 years (lowest SI) | [49] |
| *Celtis occidentalis* | SI 10 at 50 years (lowest SI) | [49] |
| *Quercus bicolor* | SI 10 at 50 years (lowest SI) | [49] |
| *Quercus palustris* | SI 10 at 50 years (lowest SI) | [49] |
| *Larix laricina* | SI 13 at 25 years [46] | [47] |
| *Populus balsamifera* L. | SI 22.5 at 50 years [41] | [44] |
| *Gleditsia triacanthos* | SI 10 at 50 years (lowest SI) | [49] |
| *Prunus serotina* | SI 10 at 50 years (lowest SI) | [49] |
| *Ostrya virginiana* | SI 10 at 50 years (lowest SI) | [49] |
| *Pinus strobus* | SI 9 at 25 years [45,46] | [45] |
| *Pinus rigida* | SI 9 at 25 years [45,46] | [45] |
| *Tsuga canadensis* | SI 9 at 25 years [45,46] | [45] |

**Table 2.** *Cont.*

| Afforestation Species | Site Index (SI) (m) | Reference for the Growth Curve |
| --- | --- | --- |
| *Quercus rubra* | SI 10 at 50 years (lowest SI) | [49] |
| *Pseudotsuga menziesii* | SI 20 at 50 years [51] | [51] |
| *Quercus velutina* | SI 10 at 50 years (lowest SI) | [49] |
| *Quercus coccinea* | SI 10 at 50 years (lowest SI) | [49] |
| *Populus deltoides* | SI 22.5 at 50 years [41] | [44] |
| *Acer nigrum* | SI 15 at 50 years | [52] |

## 3. Results

### 3.1. Roadside Afforestation Scenarios

For the Mauricie region, the simulated total ecosystem carbon stock of the plantations, 100 years after afforestation of the roadside, varied from 264 Mg C ha$^{-1}$ for the Standard scenario and 335 Mg C ha$^{-1}$ for the Limited Maintenance scenario. For the Montréal region, the total ecosystem carbon stock after 100 years varied from 265 Mg C ha$^{-1}$ for the Standard scenario and 344 Mg C ha$^{-1}$ for the Diversified scenario. Similar to Montréal, the highest ecosystem carbon stock in Montérégie after 100 years was associated with the Diversified scenario (334 Mg C ha$^{-1}$), and the lowest was found with the Standard scenario (262 Mg C ha$^{-1}$) (Figure 2).

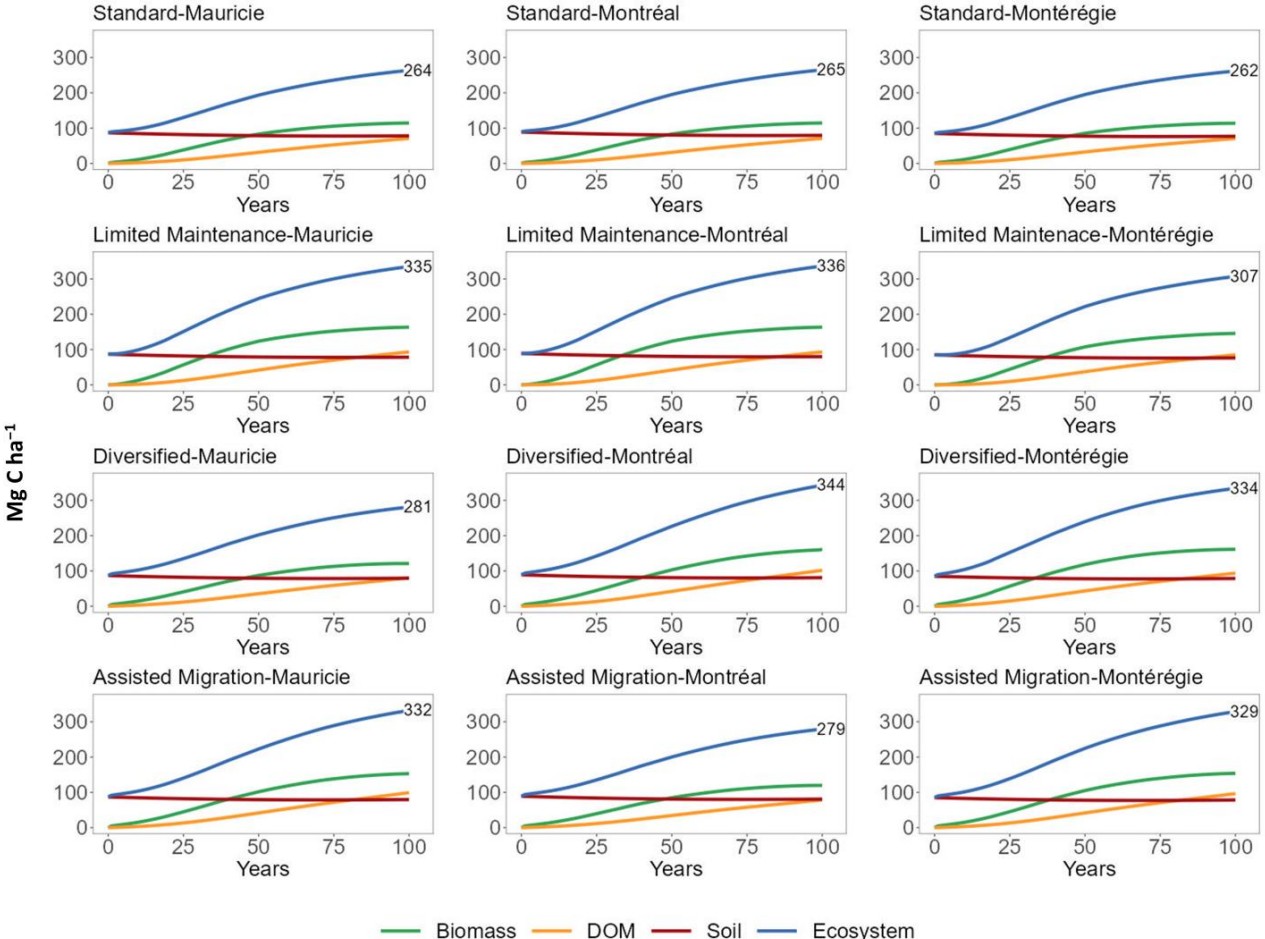

**Figure 2.** Carbon stock evolution (Mg C ha$^{-1}$) in ecosystem pools (biomass, dead organic matter or DOM, soil, and total ecosystem) for the afforestation scenarios in different study regions. The number shown on each graph is for the total stocks reached in the ecosystem at the end of the simulation.

The biomass carbon content (above- and belowground) was the largest carbon pool in all afforestation scenarios. At the end of the simulation period, the biomass carbon content ranged from 115 to 153 Mg C ha$^{-1}$ in Mauricie, with the Limited Maintenance scenario exhibiting the highest and the Standard scenario exhibiting the lowest carbon stocks. In Montréal, the Limited Maintenance scenario (163 Mg C ha$^{-1}$) had the highest carbon stocks in biomass, while the lowest was found in the Standard scenario (115 Mg C ha$^{-1}$). In Montérégie, the largest biomass carbon stock was produced by the Diversified scenario (161 Mg C ha$^{-1}$), while the smallest one was produced by the Standard scenario (114 Mg C ha$^{-1}$).

The second-largest carbon pool was aboveground dead organic matter (DOM), with a maximum amount of 102, 99, and 96 Mg C ha$^{-1}$ in Montréal (Diversified scenario), Mauricie (Assisted Migration scenario), and Montérégie (Assisted Migration scenario), respectively. The Standard scenario (71 Mg C ha$^{-1}$) had the lowest amount of carbon in the DOM pool at the end of the simulation period for all regions.

The initial soil carbon pool of the afforestation scenarios following soil ploughing (i.e., accounting for an initial loss of 10 Mg C ha$^{-1}$) was used as an input in CBM-CFS3. It was set at 87, 89, and 85 Mg C ha$^{-1}$ at the start of the simulation process for Mauricie, Montréal, and Montérégie, respectively. Based on the simulated data, the growth of the plantation was accompanied by a decrease in soil carbon storage for all scenarios. At the end of the simulation (i.e., after 100 years), the soil carbon stock decreased from 87 to 78–79 Mg C ha$^{-1}$ in Mauricie, from 89 Mg C ha$^{-1}$ to 80–81 Mg C ha$^{-1}$ in Montréal, and from 85 Mg C ha$^{-1}$ to 76–79 Mg C ha$^{-1}$ in the Montérégie region.

### 3.2. Comparison of Simulated Scenarios with Reference Roadsides

Carbon stocks from simulated plantations and from field data collected in reference roadsides were in the same range (Figure 3), confirming that CBM-CFS3 can adequately represent carbon dynamics on roadsides. Nevertheless, our simulation results showed that all afforestation scenarios had higher total carbon stocks over the simulation period than those stored in roadsides currently classified as agricultural lands and rights-of-way, i.e., sites for which natural vegetation succession is likely not actively occurring (Figure 3). However, compared with roadsides classified as abandoned agricultural land for at least 25 years, our simulations indicated that it takes approximately 10 to 15 years in Mauricie for the ecosystems resulting from the afforestation scenarios to accumulate an equivalent quantity of carbon. On the other hand, in Montérégie, reference roadsides classified as abandoned agricultural land appeared to contain less carbon than the plantations emerging from afforestation scenarios. Yet, according to the historical land-use trajectories of roadsides (Table 3), these sites may potentially evolve into forests over time.

Furthermore, our results showed that the reference roadsides currently classified as forests, some of which evolved from the abandonment of agriculture and others that were at least 45 years of age (45 years corresponding to a minimum age), actually exhibited higher carbon stocks than those projected in the afforestation scenarios for at least 60 to 84 years following plantation, depending on the region and species mix. After this period, the simulated roadside plantations would surpass the carbon stock measured in the reference roadsides. Also, our analysis indicated that the most considerable differences between the measured reference roadsides and the simulated scenarios were observed in the soil and live biomass carbon pools (Figure 2, Table 3). Indeed, we observed higher carbon stocks in the soil of the reference roadsides than in the simulated plantations resulting from afforestation scenarios; the highest carbon stock in soil was found in reference roadsides classified as forests in all three regions, and the lowest was found in abandoned agricultural land (Table 3).

**Table 3.** Evolution of land use (AL: agricultural land; AAL: abandoned agricultural land; ROW: right-of-way, forest, and urban sites) for the reference roadsides in the three study regions, with the means of current (2022) C stocks in Mg C ha$^{-1}$ in biomass (including above- and belowground pools), dead organic matter (including snags, woody debris, litter, and forest floor), soil and total ecosystem, according to the current land use.

| Study Region | Number of Reference Roadsides (n) | Historical Land Use | | | | Current Land Use (Since 2015) | Carbon Stocks (Mg ha$^{-1}$) | | | |
| | | 1975 | 1985 | 1995 | 2005 | | Live Biomass (Aboveground and BelowGround) | Dead Organic Matter (Snags Woody Debris, Litter, and Forest Floor) | Soil (Mineral Horizons from 0 to 55 cm) | Total Ecosystem |
|---|---|---|---|---|---|---|---|---|---|---|
| Mauricie | 12 | Forest | Forest | Forest | Forest | Forest | 124.21 ± 20.34 | 33.19 ± 7.00 | 111.08 ± 19.54 | 268.50 ± 30.77 |
| | 2 | AL | Forest | Forest | Forest | Forest | 143.35 ± 11.74 | 9.82 ± 0.62 | 113 ± 44.95 | 266.16 ± 33.83 |
| | 5 | AL | AL | AL/ AAL | Forest | Forest | 62.10 ± 20.19 | 2.05± 0.86 | 116.39 ± 25.77 | 180.54 ± 33.27 |
| | 2 | AL | AL | AL | AL | AL | 1.70 ± 0.66 | 2.16 ± 0.55 | 64.81 ± 7.62 | 68.66 ± 7.73 |
| | 7 | AL | AL | AAL | AAL | AAL | 34.11 ± 8.57 | 3.22 ± 1.11 | 64.08 + 4.54 | 101.41 ± 10.58 |
| Montréal | 9 | Forest | Forest | Forest | Forest | Forest | 135.54 ± 11.68 | 33.30 ± 4.38 | 115.63 ± 17.87 | 284.48 ± 31.22 |
| | 1 | AL | AAL | Forest | Forest | Forest | 103.92 | 31.84 | 92.15 | 227.91 |
| | 8 | AL | ROW/ Urban | AAL/ ROW | ROW | ROW | 0.64 ± 0.14 | 0.36 ± 0.36 | 82.16 ± 4.41 | 83.16± 4.51 |
| Montérégie | 1 | Forest | Forest | Forest | Forest | Forest | 143.24 | 44.11 | 103.07 | 290.4 |
| | 2 | AL/ AAL | Forest | Forest | Forest | Forest | 129.77 ± 14.25 | 35.05 ± 22.84 | 95.57 ± 5.90 | 260.4 ± 2.7 |
| | 1 | AL/AAL | ROW | Forest | Forest | Forest | 151.08 | 13.59 | 132.02 | 296.70 |
| | 1 | AL | AAL | Urban | Forest | Forest | 124.28 | 22.87 | 95.01 | 242.4 |
| | 1 | AL | AL | AAL | AAL | AAL | 0.67 | 0.00 | 48.34 | 49.01 |

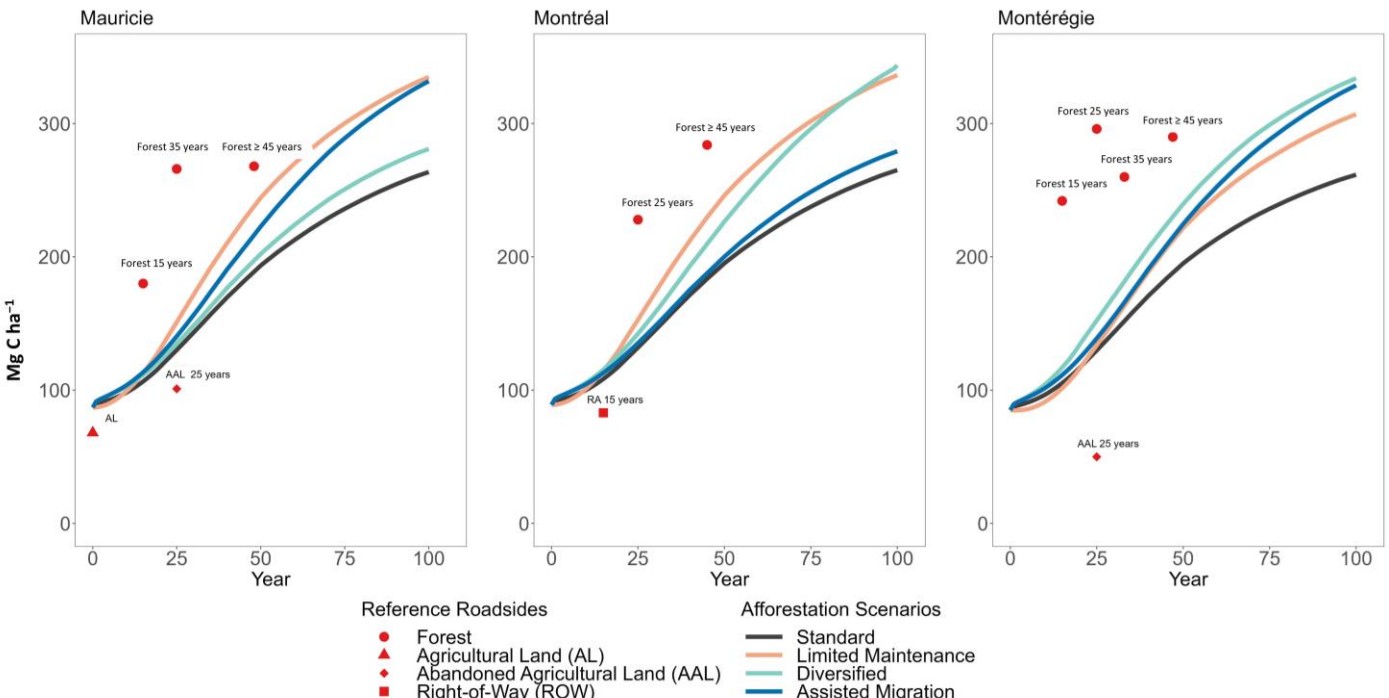

**Figure 3.** Simulated total ecosystem carbon stock (Mg C ha$^{-1}$) of the afforestation scenarios, with site measures of reference roadsides (shapes) in Mauricie, Montréal and Montérégie. The total ecosystem stocks include the carbon stock in live biomass, dead organic matter, and soil. The age of reference roadsides indicates the approximate time of agricultural abandonment based on ecoforest maps; for reference roadsides classified as Forest ≥ 45 years, 45 corresponds to the minimum age.

On the other hand, we observed higher carbon stocks in DOM (snags, woody debris, litter, and forest floor) in plantations of the simulated plantations than what was measured in the reference roadsides. The highest amounts of DOM for each region were 102, 99, and 96 Mg ha$^{-1}$ in Montréal (Diversified scenario), Mauricie (Assisted Migration scenario), and Montérégie (Assisted Migration scenario), respectively. The Standard scenario resulted in the lowest carbon stocks in the DOM pool, with only 71 Mg ha$^{-1}$ at the end of the simulation period in all the regions studied. In contrast, the highest C stocks in the DOM for the reference roadsides were found in Montérégie on sites currently classified as a forest with 44.11 Mg ha$^{-1}$, while the lowest carbon stocks in the DOM were found in abandoned agricultural lands in Montérégie and rights-of-way in Montréal (Table 3).

Increasing the growth rate of simulated species by 5% and 10% resulted in a variation in carbon stock in the ecosystem of plantations compared with the original simulated results (Figure 4). The highest-performing afforestation scenarios (Limited Maintenance in Mauricie and Diversified in Montréal and Montérégie) were found to have 9 Mg ha$^{-1}$ and 18 Mg ha$^{-1}$ more carbon stock in the total ecosystem at the end of the simulation when the growth rates were increased by 5% and 10%, respectively. In contrast, the lowest-performing scenario only showed an increase of up to 12 Mg ha$^{-1}$ in total carbon stock. Additionally, the carbon stock in live biomass increased between 4 Mg ha$^{-1}$ and 12 Mg ha$^{-1}$ depending on the level of growth increase. In comparison, the carbon stock in dead organic matter increased between 2 Mg ha$^{-1}$ and 6 Mg ha$^{-1}$ after 100 years of afforestation. However, there was no significant increase in soil carbon stock when growth was increased. Moreover, the carbon storage in the forest reference roadsides remained higher than that in the simulated plantations in all three regions for at least 5 to 6 decades, even when the plantation growth rate was increased by 5 or 10% (Figure 4).

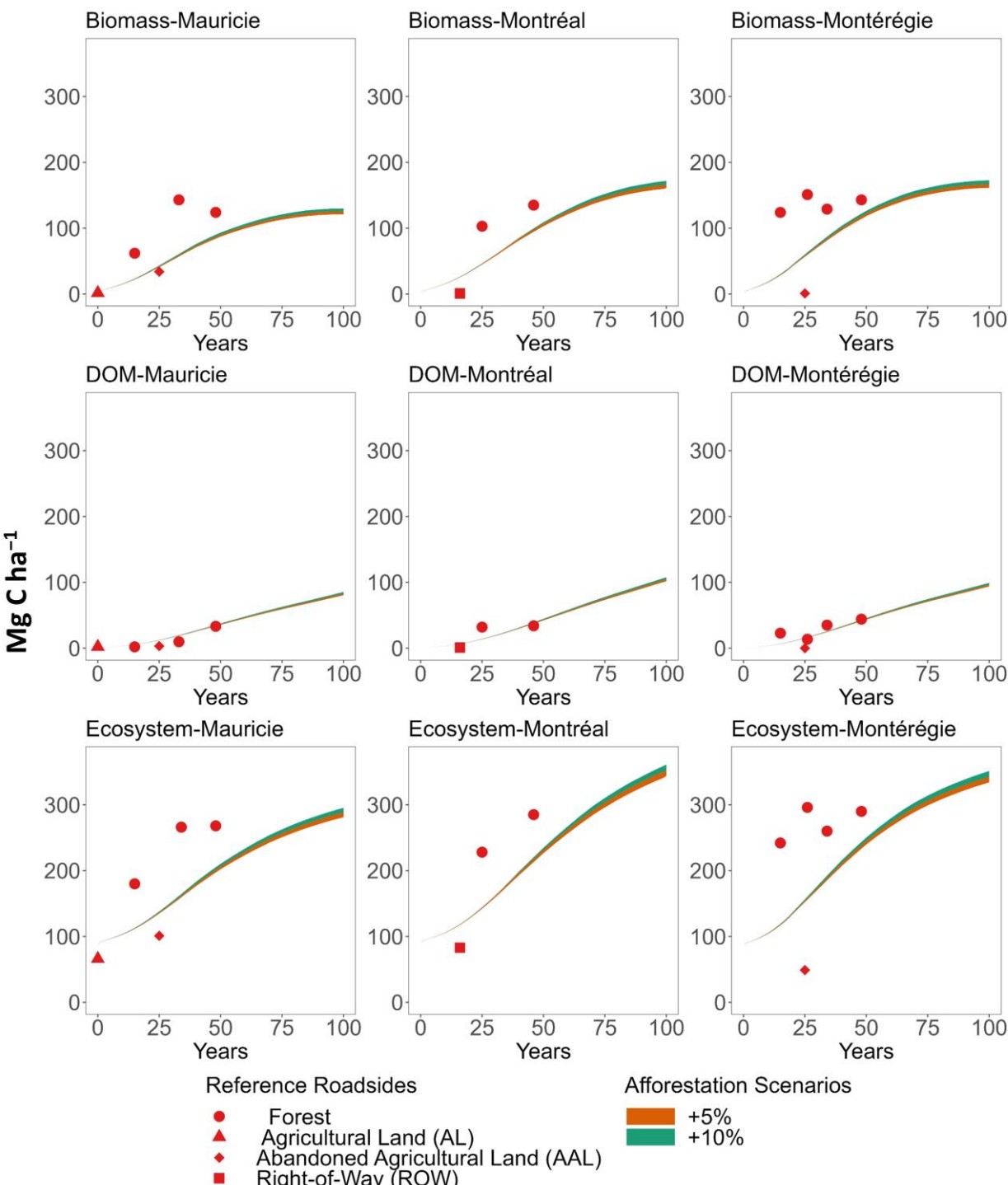

**Figure 4.** Simulated biomass, dead organic matter (DOM), and total ecosystem carbon stocks (Mg C ha$^{-1}$) of the highest-performing afforestation scenarios (Limited Maintenance in Mauricie and Diversified in Montréal and Montérégie) generated by the increase in the species growth curve (+5% and +10%), with the measures of reference roadsides (shapes) in Mauricie, Montréal, and Montérégie. Biomass includes the carbon stock in above- and belowground biomass, DOM includes the carbon stock in snags woody debris, litter and forest floor, and total ecosystem stock includes the carbon stock in live biomass, dead organic matter, and soil. For the reference roadsides classified as forest at 45 years, this corresponds to a minimum age.

## 4. Discussion

This study evaluated the GHG mitigation potential of establishing new plantations on previously unforested roadside sites in three regions of southern Québec. We simulated the evolution of carbon stocks of different mixtures of planted species based on local preferences, using regional average roadside conditions as input for the simulations. As a form of validation and evaluation of plantation performance, we then compared results with carbon stocks measured on roadsides from the same regions and under baseline conditions, i.e., in the absence of a plantation. Some of these reference roadsides have naturally evolved towards forests over the past decades, while others are still at an early stage of vegetation succession due to their history and current land use. Our simulation results suggested that the plantation of roadsides that are likely submitted to higher anthropogenic pressure and on which natural succession does not seem to be present, such as rights-of-way, can increase the carbon sink of these sites compared with a baseline scenario without active carbon management. Our findings are consistent with the research conducted by O'Sullivan et al. [53] on road verges in the UK. They showed that conserving mature trees and planting new ones can significantly improve ecosystem productivity and enhance carbon retention. A study conducted along two heavily trafficked highways in Taiwan [54] also showed that trees in these ecosystems play a significant role in carbon sequestration, with an estimated 19.9 and 12.3 kt of carbon sequestered in trunks and branches, respectively [54].

On the other hand, although our reference roadsides did not provide a real chronosequence for the evolution of carbon stocks in the absence of plantations against which to compare the simulation projections, our results suggest that roadsides classified as abandoned agricultural lands in Quebec tend to naturally evolve towards forest ecosystems and sequester significant amounts of carbon along the process of natural succession (as also observed by Thibault et al. and Tremblay and Ouimet [20,22]). This is why our simulated plantations could take up to 15 years to catch up to the carbon levels of the roadsides classified as abandoned agricultural lands and up to 84 years to catch up to the roadsides classified as naturally grown forests. When using increased tree growth rates (+10%), our simulated plantation scenarios still take several decades to reach similar carbon stocks as the reference roadsides classified as forests.

However, the tree growth rates used in the simulations might still be considered conservative, and plantations with higher yields are possible. Indeed, the field inventory of our roadside references suggested that such sites can support productive vegetation growth and carbon sequestration [15]. Moreover, it should be noted that the age of the roadside references was determined by the year of inventory when a change in land use occurred. For sites continuously classified as forests since 1975 (the oldest inventory) and that have not undergone a change in land use, we could only estimate a minimum age (i.e., 45 years); the forest may actually be much older than that and may thus have taken a long period of time to accumulate such carbon stocks. With a more predictable growth and carbon sequestration rate, it is possible that roadside plantations reach similar carbon stocks over a shorter period of time, a hypothesis only long-term field data can confirm.

Planting trees along roadsides in Bangladesh was found to play a crucial role in increasing carbon sequestration in this particular type of land [8]. However, previous land use is the most crucial factor determining the soil carbon storage associated with afforestation efforts [55], as it influences the development of present vegetation communities [56]. For example, simulations with CBM-CFS3 suggested that the afforestation of boreal open woodlands in Quebec can significantly increase the carbon sequestration potential of such boreal sites, while the plantation of abandoned agricultural lands may provide fewer convincing benefits in the context of Quebec, as they can naturally quickly revert to natural forests [5]. While abundant interest has been invested in tree planting to mitigate climate change, protecting/promoting natural succession can also be a planned strategy for augmenting carbon sinks in specific circumstances [2,57]. Our results suggested that allowing roadsides classified as abandoned agricultural lands to evolve naturally into forested land, without

human interference, can accumulate carbon possibly as efficiently, and certainly at a lower cost, as planting trees. Moreover, it would be especially inappropriate to plan plantations on roadsides currently supporting natural vegetation composed of trees and shrubs, such as the reference roadsides classified as forest; these sites could be protected, as they already represent significant carbon stocks [15].

*4.1. Vegetation Diversity and Structure*

There were important limitations for simulating plantations with diverse species mixtures with the CBM-CFS3 model. Indeed, this model can only simulate two species from each stand, i.e., one deciduous species and one coniferous species. We took steps to address this limitation and approximate carbon dynamics under mixed-species conditions. However, this did not allow us to consider the potential interactions between species within an ecosystem that may influence vegetation productivity, resilience, and carbon sequestration capacity. For instance, monoculture species exhibited a higher mortality rate than mixed plantations in a 15-year planting experiment in tropical forests [58]. Similarly, European mixed forests were found to be more resilient to drought than monospecific stands [59]. Species richness was shown to increase vegetation productivity by 17.3% compared with monocultures, increasing carbon storage both above- and belowground [60]. Similarly, a meta-analysis showed that carbon storage of young mixed species plantations was 70% higher than in monocultures [61].

Moreover, the study conducted by Srour et al. [15] examined the impact of functional diversity on carbon storage in vegetation emerging from natural succession along roadsides in Quebec. It was found that functional diversity mediated by functional dispersion was the most significant predictor of increased carbon storage, likely due to greater resource utilization and niche complementarity. While CBM-CFS3 could not capture such processes, the simulated scenarios (decided by local stakeholders) that sequestered the highest amount of ecosystem carbon after 100 years were based on species mixtures that could adapt to competitive, low-maintenance, and open environments, making them ideal for site colonization (Limited Maintenance scenario) or an abundance of long-lived broadleaved species, with conifers and other companion species (Diversified scenario). Striking a balance between local stakeholders' and land managers' needs and preferences regarding planted species and functional diversity should be achievable [62].

Another limitation of the CBM-CFS3 model is that it does not account for carbon storage in understory vegetation, such as shrubs and herbaceous species, which may develop alongside the planted trees. Although the model assumes that these types of plants do not significantly contribute to carbon storage, their omission from the model could lead to an underestimation of the carbon storage capacity of these ecosystems. Indeed, the presence of shrubs and herbaceous plants in roadside ecosystems could potentially serve as a significant carbon storage pool. It was found that shrubs can account for up to 11% of carbon storage in the roadside vegetation of Southern Quebec, with a potential stock of 15.94 Mg ha$^{-1}$, while herbaceous species can represent 9% of total carbon storage in areas where trees and herbs are dominant, with an average of 1.7 Mg ha$^{-1}$ [15]. Similarly, on a heavily trafficked expressway in China, shrubs and herbaceous plants contributed 24% of the vegetation carbon storage [6].

Yet, due to a lack of precise data on the usage and maintenance history of our studied roadsides, we can only make assumptions about the actual evolution of vegetation and the drivers of this evolution. For instance, changes in the use of certain roadsides over the past few decades (i.e., due to the abandonment of agriculture) may have led to different stages of natural succession in the vegetation of these sites. Additionally, some roadsides (such as sites classified as rights-of-way) may have undergone regular mechanical pruning of vegetation, which can interrupt the natural progression of succession.

Since the ecological conditions used as input for the simulations were based on field data collected on reference roadsides, the latter could be considered valid natural comparison points for the simulated plantations. However, several factors can contribute

to the higher carbon stocks found in natural vegetation in reference roadsides compared with simulated plantations. One possible explanation is that natural vegetation tends to be more diversified and vertically stratified (trees, shrubs, and herbs), which could have led to higher productivity in the reference roadsides despite a high diversity of tree species in plantation scenarios. Indeed, species structure has been found to impact significantly forest productivity compared with species diversity [63,64]. Another factor that could account for the estimated differences in carbon stocks between plantations and natural vegetation is the spatial distribution of species [65]. The random distribution of stands in natural vegetation tends to promote greater vegetation and development than the uniform distribution found in plantations. Additionally, we observed that natural vegetation often includes species with varying diameters, which can explain the differences in vegetation sizes seen in natural roadside references. Finally, stand density is another factor that can impact forest productivity, with an optimal density allowing for the equal distribution of resources between species [66] and decreased competition, which can improve overall productivity and carbon sequestration. By managing plantations with complex structures [67] and optimizing their spatial distribution [65] based on natural vegetation patterns, it may be possible to enhance ecosystem functioning, stand productivity, and carbon storage. Improving site conditions, such as soil properties, could also boost plantation productivity, as could identifying the ideal location for planting based on natural vegetation patterns and site conditions.

*4.2. Soil Carbon Stocks*

Our simulation results suggested that the afforestation of roadsides caused a decrease in soil carbon storage over the simulation period. A similar reduction in soil carbon stocks was obtained by Fradette et al. [19], who simulated the afforestation of abandoned agricultural lands in Quebec; this decrease was attributed to the difference in input and output of organic matter. In our case, this decrease was in addition to the initial soil carbon loss due to ploughing, although the limited number of collected field samples in our study did not allow for a solid analysis of the impact of ploughing on soil carbon stocks. A meta-analysis compiling data from 204 sites reported an initial decrease in soil C content during the first decades after the afforestation of agricultural lands, followed by a recovery by year 30 [68]; the study further indicated that the extent of the decrease was influenced by the previous agricultural practices and by the species used for plantation. Soil preparation prior to the afforestation of grasslands has also been shown to decrease soil carbon storage [69,70]. Yet, Tremblay et al. [71] found that carbon storage in soil decreased in the first 22 years following the afforestation of abandoned agricultural land in Quebec, which was attributed to a lower litterfall rate in the first years rather than to soil preparation. Whether the decrease in soil carbon stocks observed in our simulations—likely due to a combination of carbon loss following site preparation and an insufficient organic matter input relative to soil decomposition—would actually occur on the field remains to be seen. It is important to note that soil carbon accumulation occurs gradually and spans multiple years as decaying organic matter is integrated into the soil. This may also contribute to the decrease in soil carbon storage observed in our afforestation results.

**5. Conclusions**

Our study highlighted the potential of the afforestation of roadsides for increasing carbon sequestration, using southern Québec as a case study. The potential benefits of roadside plantations were highly dependent on the assumptions made regarding the reference roadsides used in this study. These assumptions were based on field and cartographic data and cannot replace systematic monitoring and documentation of control and plantation sites over time. Only long-term monitoring of the plots used in this study will make it possible to precisely establish the comparative performance of the reference and plantation scenarios.

Our results suggested that plantations on roadsides subjected to higher anthropogenic pressure, such as road rights-of-way, should provide carbon sequestration benefits relative

to a baseline scenario (i.e., without plantation); indeed, planting trees would result in a significant increase in C compared with their current herbaceous stage. On the other hand, allowing roadsides classified as abandoned agricultural lands to undergo natural succession, which, in the context of southern Quebec, appears likely to lead to forest vegetation, may effectively promote carbon storage on roadsides without the need for plantation.

Thus, the promotion of natural vegetation on roadsides that were used for agriculture in the past but have otherwise not been subjected to intense anthropogenic degradation can create a significant sink of carbon, which, as a low-cost nature-based solution, limits the interest in tree plantation on these sites. Nevertheless, planting trees to accompany natural vegetation can increase the density and functional diversity of existing vegetation, potentially improving their carbon accumulation potential [15]. Plantation on degraded lands such as rights-of-way and urban areas can create a new forest cover and increase carbon sequestration, even when accounting for the impact of site preparation on soil carbon stocks. Implementing these approaches can significantly contribute to carbon sequestration and aid efforts to mitigate climate change.

The lack of natural vegetation evolution that we observed on degraded sites might have been caused by intensive mechanical vegetation control and not by intrinsically hostile site characteristics. The data collected during this project cannot provide a conclusive answer. Yet, minimizing mechanical or chemical maintenance of vegetation on roadsides could be a low-cost option considering the potential capacity for carbon sequestration of natural vegetation.

Despite its limitations, CBM-CFS3 still plays an essential role in simulating the effect of forest management practices such as afforestation on carbon storage. Nevertheless, it will be necessary to properly document and track carbon storage in control and planted plots on roadsides to validate long-term projections. Establishing an accurate baseline scenario can especially provide valuable information on carbon loss and gain after afforestation practices.

**Author Contributions:** Conceptualization, N.S.; methodology, N.S.; software, N.S.; validation, E.T. and J.-F.B.; formal analysis, N.S.; investigation, N.S. and E.T.; resources, E.T.; data curation, N.S.; writing—original draft preparation, N.S.; writing—review and editing, N.S., E.T. and J.-F.B.; visualization, N.S.; supervision, E.T. and J.-F.B.; project administration, E.T.; funding acquisition, E.T. All authors have read and agreed to the published version of the manuscript.

**Funding:** This research was funded by the Quebec Ministry of Transportation (Research project R 833.1; PI: E. Thiffault).

**Data Availability Statement:** The data of this study are available at http://doi.org/10.6084/m9 .figshare.24881472 (accessed on 29 January 2024).

**Acknowledgments:** The authors thank the Forestry Education and Research Center of Sainte-Foy (CERFO) for site identification and documentation; and the members of the Team Carbone. Special thanks to David Rivest and Alain Paquette for their contribution to the project.

**Conflicts of Interest:** The authors declare no conflicts of interest.

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
