# Peer review of "Exploring the Potential of Roadside Plantation for Carbon Sequestration Using Simulation in Southern Quebec, Canada"

_forests, doi:10.3390/f15020264_

Round 1
Reviewer 1 Report
Comments and Suggestions for Authors
· Some of the references are in different fonts. They are highlighted in the annotated manuscript. Please change them.
· Table 1: Inset a common column for three locations.
· Line 178: give the full form at first mention
· Line 193: add "elemental analyzer"
· Line 198: check the punctuation mark
· Please correct the superscripts (lines 220-221)
· Line 230: check “th”, may be a typo
· Please check the reference style (line 333)

It is of good standard.
Reviewer 2 Report
Comments and Suggestions for Authors
Dear Authors,
The subject of the manuscript entitled “Exploring the potential of roadside afforestation for carbon sequestration: A comparison of simulated plantations with natural vegetation in southern Quebec, Canada.” fit the profile of “Forests” journal. The article is well arranged and well written, indicating its importance of roadside vegetation in mitigation of current climate change and global warming issues. The findings of this article will be interesting to national or international community so the article get published in this journal.
Abstract
Kindly check minor spelling mistake. The abstract needs a better composition of words. Moreover, I would suggest to mention some more important values (Results) in this section.
Introduction
Introduction is well documented. However, research gaps at some places are not completely described in this section. The authors only focused on urban green spaces not the parks and roadside woody vegetation. I would suggest to add some material regarding the worldwide trend of carbon sequestration in roadside vegetation to make it more interesting for the readers.
Material and methods
Materials and Method section is well explained.
Results
Results are well indicated. Pay special attention to grammar and spelling mistakes.
Discussion
This portion needs improvements. The authors must some lines which should be more focused for highlighting the important finding of this work and also compare the findings with previous studies for better looking discussion. Relevant tables and figures must be cited in discussion section also. This will facilitate the reader to follow the flow of the article.
References
Check and format the references according to the journal format
Conclusion
Minor language issues must be addressed to improve quality of the MS
Reviewer 3 Report
Comments and Suggestions for Authors
My comments:
This manuscript, entitled “Exploring the potential of roadside afforestation for carbon sequestration: A comparison of simulated plantations with natural vegetation in southern Quebec, Canada”, aimed to examine the potential for carbon sequestration and storage of plantations on roadsides in southern Quebec, Canada. The author used a model (Canadian Forestry Department 3 Carbon Budget Model, CBM-CFS3) to predict the carbon storage dynamics of three study sites under different plantation scenarios. Meanwhile, the author selected land parcels as reference roadsides in the three studied areas to investigate the carbon storage based on different land use histories. The results of modeling and field investigations showed that roadside plantations can provide carbon sequestration benefits relative to baseline conditions. However, if the roadside was abandoned farmland and underwent natural succession, they may obtain higher carbon storage than planting.
The theme of this manuscript is important and interesting. The method is reasonable. This manuscript is likely well organized. The content of the manuscript is well within the scope of publication of “Forest”. However, this manuscript has some flaws and cannot be published in the current version. The main issue is that the "Methods" section did not provide sufficient information and the number of the experimental plot is too small. Here, I would like to elaborate on my views as follows:
Lines 18-22 (L18-22), it would be better if the authors could provide quantitative data for the results, such as the quantitative data for ‘carbon sequestration benefits relative to baseline conditions’ and “undergo natural succession could promote higher carbon storage on roadsides than planting”.
L40-41, I would like to see some concrete results or findings as examples about “Lands alongside road networks are under-utilized areas that could contribute to increased carbon storage through afforestation”.
L42-73, these two paragraphs may be combined as one paragraph.
L84-87, 2.1. Study area. more information are required about the geographical coordinates, climate, geography, topography, soil, vegetation etc. in the study area.
L89, a new map should be provided. From the current map, I have no idea about this study area and country; where are Mauricie, Montréal and Montérégie in the map?
L90, what is the experimental design for this study? One section, “Experimental design” should be provided.
L94-95, “different sectors of the three studied regions (Figure 1)”. How many sectors were set up in this study and distributed in each of the three studied regions?
L100-101, “one inventory plot established on each distinct roadside”. How many distinct roadside types are there in this study? What is the size of the plot? From a statistical perspective, the number of plots (one parcel per roadside type) is too small to represent the entire roadside type.
L107, How many roadsides were selected in this study? “60 selected roadsides” in L107, but “Fifty-two roadsides” in L98-99.
L113, How many “main plots” were there in each roadside selected in this study?
L117, How many subplot “(a plot with a 3.98 m radius (50 m²))” were there for each main plot?
L120-121, How long is the transaction line for the “two perpendicular transects”?
L121, How many classes are there for the “decay class” of woody debris and what is the definition about them?
L171-173, Table 1 needs to be reconstructed. The species could be represented by the number. No all species are showed up and show the main tree species and their %. What is the stand density for each of the Afforestation scenarios?
L175, the main structure of the CBM-CFS3 model should be provided. What are the mail input data and output data for this model.
L188, how is the size of the “A treated plot”? If the size of this “A treated plot” does not mach the plot size (L176: 1 hectare of roadside ) used in the CBM-CFS3 model, how did the authors treat the differences of the two resolutions?
L188-190, The content of this sentence is very confusing and needs to be rewritten.
L214, one section of “Data Analysis’ should be added. What method did the authors use to evaluate the results estimated by the CBM-CFS3 model?
L230, this sentence is not completed.
L263, In Figure 3, the actual age of “Forest 45 years” might excess 45 years.
L397, why did the soil carbon stock simulated by the CBM-CFS3 model consistently decline over 100 years? Soil ploughing might only affect soil carbon stock at early stage in vegetation succession. As the total biomass carbon content considerably increased, the return of belowground biomass carbon should cause an increase of carbon in the soil.
Reviewer 4 Report
Comments and Suggestions for Authors
This reviewer has difficulties in coping with this paper.
It appears from Figure 3 that during the first 50 years, the accumulated carbon stock is greater in non-afforested sites than in afforested sites. This reviewer can find at least two possible reasons:
1. The empirical data from non-afforested sites and the simulation of afforestation results are not comparable.
2. The afforestation procedures impair the productive capacity of the sites.
,Explanation (1) appears more likely for this reviewer. This explanation possibly nullifies the content of the paper.
If explanation (2) is valid, this should be discussed carefully. What is wrong with the procedures that impair the productive capacity of the sites? What kind of procedures would avoid such impairment?
The Authors possibly should very carefully verify that the presented conclusions are based on results. On the first reading, this reviewer was unable to comprehend the justification of conclusions regarding the effect of the anthropogenic pressure.
Round 2
Reviewer 4 Report
Comments and Suggestions for Authors
The simulation results appear not to be comparable with the field observations. This apparently nullifies most of the paper content.
